# Identification of a non-axisymmetric mode in laboratory experiments searching for standard magnetorotational instability

Yin Wang [1] ✉, Erik P. Gilson [1], Fatima Ebrahimi[1,2], Jeremy Goodman [2], Kyle J. Caspary[1], Himawan W. Winarto[2] & Hantao Ji [1,2]

The standard magnetorotational instability (SMRI) is a promising mechanism for turbulence and rapid accretion in astrophysical disks. It is a magnetohydrodynamic (MHD) instability that destabilizes otherwise hydrodynamically stable disk flow. Due to its microscopic nature at astronomical distances and stringent requirements in laboratory experiments, SMRI has remained unconfirmed since its proposal, despite its astrophysical importance. Here we report a nonaxisymmetric MHD instability in a modified Taylor-Couette experiment. To search for SMRI, a uniform magnetic field is imposed along the rotation axis of a swirling liquid-metal flow. The instability initially grows exponentially, becoming prominent only for sufficient flow shear and moderate magnetic field. These conditions for instability are qualitatively consistent with SMRI, but at magnetic Reynolds numbers below the predictions of linear analyses with periodic axial boundaries. Three-dimensional numerical simulations, however, reproduce the observed instability, indicating that it grows linearly from the primary axisymmetric flow modified by the applied magnetic field.

Astronomical accretion disks consist of gas or plasma orbiting a compact massive object such as a black hole or protostar, and slowly spiraling inward (accreting) by surrendering orbital angular momentum to other material in the disk or in an outflow[1]. Driven by gravity, the angular velocity profile in a Keplerian flow has a decaying power-law dependence on the cylindrical radius, $\Omega(r) \propto r^{-q}$, with $q = 3/2$. According to Rayleigh's criterion[2], purely hydrodynamic rotation profiles with $0 < q < 2$ ("quasi-Keplerian") are linearly stable to axisymmetric perturbations, and apparently linearly and nonlinearly stable nonaxisymmetrically as well[3,4], at least without complications such as thermal effects or interactions with dust[5]. Therefore, hydrodynamic modes cannot excite the turbulence required to explain rapid accretion[6,7]. The standard magnetorotational instability (SMRI)—a unique magnetohydrodynamic (MHD) instability in a conducting Keplerian flow in the presence of an axial magnetic field—is thus regarded as one of the most promising mechanisms for unraveling the

origin of turbulence in accretion disks[3,8,9], apart from possible rapid accretion due to laminar magnetized winds[10]. Unlike the SMRI that requires only the magnetic field parallel to the rotation axis, other versions of MRI involving azimuthal fields have been found and experimentally demonstrated: helical MRI (HMRI) and azimuthal MRI (AMRI)[11,12]. While their existence is intriguing, these instabilities are inductionless, incapable of generating and sustaining the needed magnetic field. They also require steeper-than-Keplerian rotation profiles ($q > 3/2$), hence are unlikely to be relevant to most astrophysical disks[13].

Unlike other fundamental plasma processes such as Alfvén waves[14–16] and magnetic reconnection[17–19] which have been detected and studied in space and in the laboratory, SMRI remains unconfirmed long after its proposal[8,20,21] other than its analogs[22–25], despite its widespread applications in modeling including recent black hole imaging[26]. Due to its microscopic nature and the limitations of

[1]Princeton Plasma Physics Laboratory, Princeton University, Princeton, NJ 08543, USA. [2]Department of Astrophysical Sciences, Princeton University, Princeton, NJ 08544, USA. ✉e-mail: ywang3@pppl.gov

telescope resolution, SMRI cannot be captured by current astronomical observations. SMRI is also proposed[27,28] to be realized in a terrestrial Taylor–Couette experiment, which consists of two independently rotating coaxial cylinders that viscously drive the liquid metal between them to a quasi-Keplerian flow with a magnetic Reynolds number larger than unity. However, as the axial boundaries (endcaps) of a conventional Taylor–Couette cell are bound either to the inner cylinder or to the outer cylinder, their motions do not match the viscously driven flow profile in the bulk. Ekman circulation is thus excited, entailing $\partial\Omega/\partial z \neq 0$ along the axial $z$-direction and some turbulence that prevent the detection of SMRI[29,30].

Here, we report a laboratory experiment searching for SMRI using a specially designed Taylor–Couette cell. The cell's copper-made endcaps can rotate independently[31], which provide a quiescent quasi-Keplerian flow in the bulk region despite that the shear Reynolds number exceeds a million[3,4,32], as well as their inductive coupling with the fluid[33–36]. Through magnetic field measurements, we identify a global MHD instability occupying the entire bulk region, which exists only at sufficiently large rotation rates and intermediate magnetic field strengths, in line with typical requirements for SMRI from linear theories. The instability is nonaxisymmetric with a dominant $m = 1$ mode in the azimuthal direction, which spontaneously breaks the rotational symmetry possessed by the system. Our numerical simulations reproduce the experimentally observed instability and further reveal that it develops from an axisymmetric base flow modified by the applied magnetic field.

## Results

We denote the radius of the inner and outer cylinders as $r_1 = 7.06$ cm and $r_2 = 20.3$ cm, and their height as $H = 28$ cm. The aspect ratio of the cell is thus $\Gamma \equiv H/(r_2 - r_1) \simeq 2.1$, which is deliberately designed to be large to ensure the magnetic diffusion time is longer than the rotation period and the Alfvén crossing time, and thus help excite SMRI[27,28]. The upper and lower sealing endcaps are split into two rings at $r_3 \simeq (r_1 + r_2)/2$. The angular velocities of the inner cylinder, inner rings, outer rings and outer cylinder are, respectively, $\Omega_1$, $\Omega_3$, $\Omega_2$ and $\Omega_2$. The corresponding frequencies and their differences are denoted as $f_i \equiv \Omega_i/2\pi$ and $f_{ij} \equiv (\Omega_i - \Omega_j)/2\pi$, respectively. For all results described here, $\Omega_1 : \Omega_2 : \Omega_3 = 1 : 0.19 : 0.58$, generating a shear flow similar to what was studied previously[36]. A set of six copper coils provides a uniform axial magnetic field $B_z$. The local radial magnetic field $B_r(t)$ is measured by Hall probes installed on the surface of the inner cylinder at various azimuths and heights. Dimensionless measures of rotation and field strength are the magnetic Reynolds number Rm $= r_1^2 \Omega_1/\eta$ and the Lehnert number $B_0 = B_z/(r_1\Omega_1\sqrt{\mu_0\rho})$, which are varied in the ranges $0.5 \lesssim R_m \lesssim 4.5$ and $0.05 \lesssim B_0 \lesssim 1.2$, respectively. The magnetic Prandtl number is $P_m = \nu/\eta = 1.2 \times 10^{-6}$ for the working fluid GaInSn eutectic alloy (galinstan). Here, $\mu_0$ is the vacuum permeability and $\nu$, $\eta$, and $\rho$ are, respectively, the kinematic viscosity, magnetic diffusivity and density of galinstan. The device spins for 2 min (several Ekman times) to ensure a relaxed hydrodynamic flow before the introduction of $B_z$.

A representative time series of the imposed magnetic field $B_z(t)$ is shown in Fig. 1a, increasing from zero to 2100 G in less than one second. The corresponding $B_r(t)$ measured in the midplane is shown in Fig. 1b, which first increases synchronously with $B_z(t)$, then saturates to a statistically stationary state. An intriguing finding is a strong oscillation in $B_r(t)$, which emerges at $t \simeq 125.6$ s and saturates after $t \simeq 127.0$ s. Such an instability appears to be global as it is well correlated at different heights. Figure 1c shows the spectrogram of $B_r(t)$ using a one-second moving window, where all frequencies are shifted by $-f_1$ relative to the lab frame. The transient power at low frequency ($f \lesssim 5$ Hz) is believed to be due to a modification of the base flow caused by the imposed magnetic field. As indicated by the horizontal lines, energy of the instability appears between the machine-induced

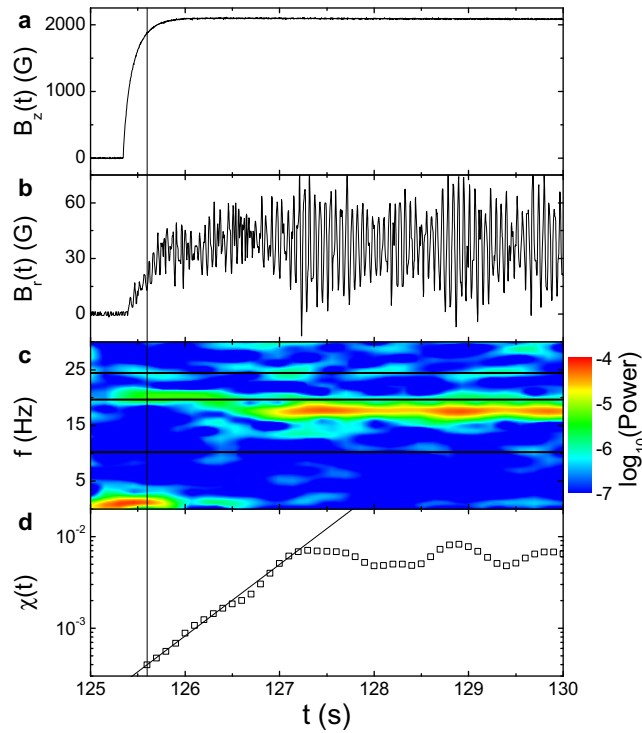

**Fig. 1 | Characterization of the instability. a** Time series of the imposed magnetic field $B_z(t)$ at Rm = 3 and $B_0 = 0.2$. **b–d** Corresponding time series of the measured radial magnetic field $B_r(t)$ (**b**), the spectrogram of $B_r(t)$ (**c**) and the instability amplitude $\chi(t)$ (**d**) in the midplane. The horizontal lines from top to bottom in (**c**) represent $f_1$, $f_{12}$, and $f_{13}$, respectively. The solid line in (**d**) shows an exponential fit, $\chi(t) \sim \exp(\gamma t)$ to the data points with $\gamma = 1.8$ s$^{-1}$. The vertical line through the four panels indicates the time when the instability starts to grow.

frequencies $f_{12}$ and $f_{13}$, and has a maximum value at $f \simeq 17.5$ Hz. This is a general feature of the instability at different Rm and $B_0$, namely, its frequency is between the rotation frequencies of the inner rings and outer cylinder. We then define the normalized strength of the instability as

$$\chi(t) = \left[\int_{1.05f_{13}}^{0.95f_{12}} P_B(f)\mathrm{d}f\right]^{\frac{1}{2}} / \langle B_z(t)\rangle, \tag{1}$$

where $P_B(f)$ and $\langle B_z(t)\rangle$ are, respectively, the power spectrum of $B_r(t)$ and the mean value of $B_z(t)$ sampled in a moving one-second window. As shown by the vertical line in Fig. 1d, the measured $\chi(t)$ starts to grow about 0.3 s after the imposition of $B_z$, and then saturates. Such a growth-saturation process agrees well with the evolution of $B_r(t)$ shown in Fig. 1b. The initial growth of $\chi(t)$ is well described by an exponential, indicating linear instability[27,28].

We also conduct three-dimensional (3D) numerical simulations using the open-source SFEMaNS code, which solves coupled Maxwell and Navier–Stokes equations using spectral and finite-element methods[37]. It contains 32 Fourier modes in the azimuthal direction. Similar to the experiment, the entire simulation process consists of two stages. In the first hydrodynamic stage, simulations are run to reach a relaxed hydrodynamic state without the external magnetic field, starting from an initial piecewise solid-body condition that follows the angular speed of the endcaps and two cylinders. In the second MHD stage, the external magnetic field is imposed and lasts until a saturated MHD state is achieved. The main difference between simulation and experiment is the viscous Reynolds number Re $= r_1^2\Omega_1/\nu$, which is on the order of $10^6$ in the experiment but only 1000 in our simulations. The relatively large viscosity (low Re) in simulation gives

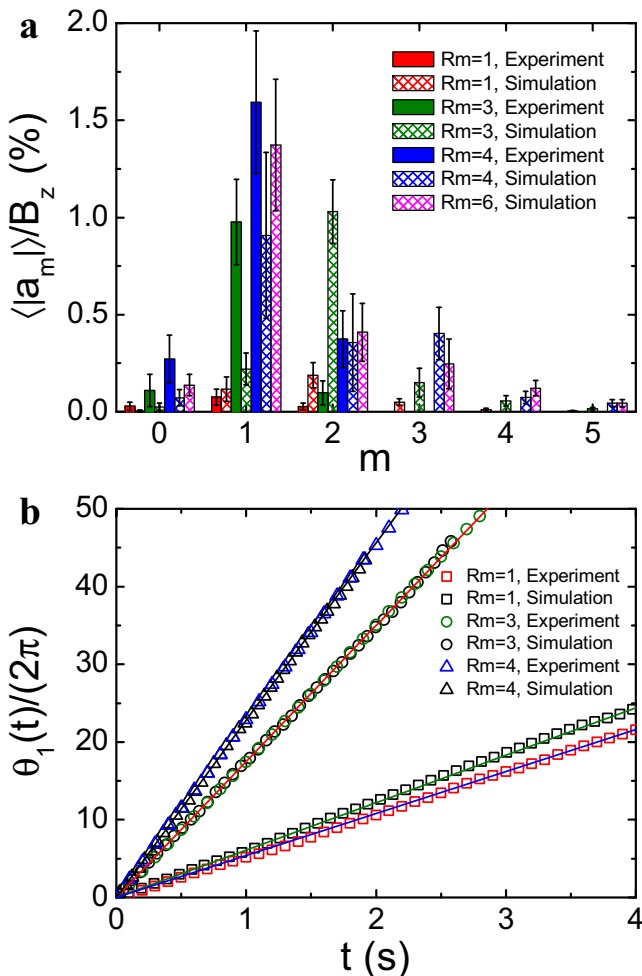

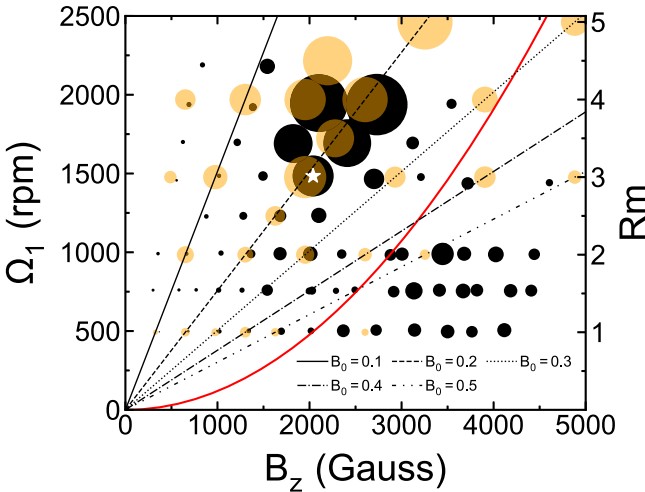

**Fig. 3 | Shear and magnetic field dependence of the instability.** Bubble plot of the instability strength from experiments (black bubbles) and 3D simulations (orange bubbles) in the $\Omega_1$-$B_z$ plane with Rm shown on the right. The data are obtained in the midplane and at the inner cylinder surface. The red curve represents the boundary for SSL instability. The straight lines show the contours of a constant $B_0$. The black bubble under the white star corresponds to the case shown in Fig. 1.

**Fig. 2 | Azimuthal structure of the instability. a** Normalized mode amplitudes $\langle|a_m|\rangle/B_z$ in the experiment (solid bars) and simulation (crosshatching bars), as a function of azimuthal mode number $m$ for different values of Rm with a fixed $B_0 = 0.2$. Measurements were made in the midplane of the inner cylinder. Error bars show the standard deviation. **b** Time evolution of the phase $\theta_1(t)/(2\pi)$ of the $m = 1$ mode in the corotating frame of the inner cylinder. The lines show a linear fit, $\theta_1(t)/(2\pi) = f_0 t$, to the data points with $f_0 = 22.8$ Hz (black), $f_0 = 17.5$ Hz (red), $f_0 = 6.1$ Hz (green) and $f_0 = 5.4$ Hz (blue).

rise to thick residual Ekman layers at the endcaps, which drive non-axisymmetric hydrodynamic modes in the bulk (see "Flow characterization in 3D simulation" in "Methods"). It was found numerically that these modes' amplitudes decrease with increasing Re[38,39], so they are undetected in the experiment. In the simulated radial magnetic field, we also observe instability similar to that in the experiment. The instability occupies the whole radial extent and rotates as a solid body (see Supplementary Movie 1), which is distinct from the Stewartson–Shercliff layer (SSL) instability, as the latter concentrates near $r \simeq r_3$ and has a spiral structure[35,40–42]. Due to the residual hydrodynamic modes, our 3D simulation cannot always reproduce the exponential growth of the instability seen in the experiment. Nonetheless, when only a single $m = 1$ mode is allowed in an otherwise two-dimensional (2D) simulation, an exponential growth is observed with a growth rate behaving similar to that in the experiment (see below).

The normalized amplitude of the $m$th azimuthal Fourier mode $\langle|a_m|\rangle/B_z$ averaged in the saturated MHD state are shown in Fig. 2a. For Rm = 1, all the $\langle|a_m|\rangle/B_z$ from the experiment are negligibly small and thus the instability is absent. The $\langle|a_2|\rangle/B_z$ at Rm = 1 from simulation is believed to be due to the residual hydrodynamic modes at low Re[38,39]. For Rm $\gtrsim 3$, the experimental $\langle|a_1|\rangle/B_z$ dominates, indicating that the

instability is nonaxisymmetric and mainly at $m = 1$. Similar results are also obtained from simulations, namely, the nonaxisymmetric amplitudes increase for Rm $\gtrsim 3$, and $m = 1$ dominates for Rm $\gtrsim 4$. At the onset of the instability (Rm = 3) in simulation, the $m = 2$ mode is the strongest, which could result from the interaction between the instability and the residual hydrodynamic modes.

Figure 2b shows a comparison between the phase $\theta_1(t)$ of the $m = 1$ mode from experiment and simulation in the saturated MHD state. All $\theta_1(t)$ are well described by linear functions, $\theta_1(t)/(2\pi) = f_0 t$ (color-coded lines), where $f_0$ is the characteristic frequency of the $m = 1$ mode. At Rm = 1, $f_0 = 5.4$ Hz in the experiment is different from $f_0 = 6.1$ Hz in the simulation. This discrepancy is caused by the Re difference between experiment and simulation, which gives rise to different hydrodynamic modes that are the main contributor to the $m = 1$ mode at low Rm. For Rm $\gtrsim 3$, on the other hand, $f_0$ from experiment agrees well with that from simulation, suggesting that the frequency of the instability is insensitive to Re. The frequency $f_0 = 17.5$ Hz for Rm = 3 from experiment is also consistent with the characteristic frequency of the instability seen in Fig. 1c.

A "bubble plot" of the instability strength in the $\Omega_1$-$B_z$ plane is shown in Fig. 3. The diameter of the experimental bubbles (black) denotes $\chi$ from Eq. (1) with a typical time window of 10 s in the saturated MHD state while their numerical counterpart (orange) follows the time average of the standard deviation of $B_r$ among azimuths. Overall agreement between experiment and simulation is excellent, namely, the instability becomes particularly pronounced only for $\Omega_1 \gtrsim 1500$ rpm (Rm $\gtrsim 3$) and $1800 \lesssim B_z \lesssim 2800$ G. The red curve shows the boundary for SSL instability in the midplane, which is stable on its left and unstable on its right. The SSL instability is nonaxisymmetric and develops from a free SSL spanning the entire vertical extent around $r \simeq r_3$[34,40,41] (see "Characterization of the Stewartson-Shercliff layer instability" in "Methods"). In particular, the SSL instability is inductionless and thus can be excited in the limit of small Rm. For example, for $\Omega_1 \lesssim 1000$ rpm (Rm $\lesssim 2$), larger bubbles appear on the right of the red curve, indicating that the SSL instability is excited. On the other hand, the large bubbles in the upper middle area are far to the left of the red curve and require $\Omega_1 \gtrsim 1500$ rpm (Rm $\gtrsim 3$), implying that the identified instability is distinct from the SSL instability. In the presence of a weak magnetic field ($B_z \lesssim 1200$ G), the bubbles from simulation are larger than their experimental counterpart. We believe

that this is due to the residual hydrodynamic modes in these simulations, for which a weak field acts as a passive tracer.

## Discussion

The above measurements reveal a global MHD instability in a modified liquid-metal Taylor–Couette experiment. The instability grows exponentially and becomes particularly pronounced once the radial shear rate is sufficiently large and the imposed axial magnetic field is moderate, consistent with typical requirements for SMRI in this system[27,28,36,43]. On the other hand, it is nonaxisymmetric with a dominant $m = 1$ azimuthal structure, which contradicts the prediction of linear theories for SMRI with an axial magnetic field in an ideal Couette flow between infinitely long cylinders: the SMRI should be axisymmetric at onset[27,28]. The instability also has minimum requirements for rotation ($\Omega_1 \gtrsim 1500$ rpm) and magnetic field ($B_z \gtrsim 1800$ G) smaller than predictions for SMRI based on local Wentzel–Kramers–Brillouin (WKB) analysis or global linear calculation (see "Linear theory predictions of SMRI" in "Methods"), implying that either it is not SMRI or the linear theories are not capable enough to describe our system without including its closed geometry and complex boundary conditions. Experiment and simulation generally agree as to the characteristic frequency, azimuthal structure, and distribution of amplitudes in the $\Omega_1 - B_z$ plane. The instability is distinct from the hydrodynamic Rayleigh instability (see "Characterization of hydrodynamic Rayleigh instability" in "Methods") and the SSL instability, which are often present and sometimes mistaken for SMRI in previous experiments[44,45]. Even with conductive endcaps, our 3D simulation nonetheless shows that the induced azimuthal magnetic field $B_\phi$ is still less than 15% of the applied axial magnetic field (see "Azimuthal magnetic field" in "Methods"). Furthermore, it is found that most $B_\phi$ is mainly concentrated in the region near the endcaps rather than the bulk region where the observed instability is located. As a result, the instability is also unlikely to be the nonaxisymmetric AMRI or HMRI, which require a pure or predominant azimuthal magnetic field[46].

While we are currently unable to pinpoint the fundamental cause for the observed instability, our "2-mode" simulations nonetheless provide a phenomenological description of its generation mechanism: it develops from an axisymmetric base flow modified by the applied magnetic field. In these simulations, only $m = 0$ and $m = 1$ modes are kept, and all the $m \geq 2$-mode amplitudes are set to zero. In the first hydrodynamic stage, a 2D flow is relaxed to a steady state from an initial piecewise solid-body rotation, where the $m = 0$ mode can evolve freely and the amplitude of the $m = 1$ mode is set to a negligibly small value ($\sim 10^{-20}$) at each time step. When an axial magnetic field is applied in the second MHD stage, both $m = 0$ and $m = 1$ modes can evolve freely.

Figure 4a shows the time series of the dimensionless amplitude $A_0$ of the $m = 0$ mode in the velocity field from the 2-mode simulations. As separated by the vertical line, $A_0$ (black curve) saturates at the end of the first stage at $t\Omega_1 = 400$, indicating that a hydrodynamic steady state is reached. When the axial field is turned on ($B_0 > 0$), there is a transient variation in $A_0$, followed by a new MHD steady state. The red curve shows a continuation of the hydrodynamic evolution ($B_0 = 0$) for comparison. Because the amplitude of the $m = 1$ mode is very small (see Fig. 4b) in the simulation time span, there is no coupling between the two modes in the second stage. The initial rapid variation of $A_0$ after the imposition of magnetic field is caused by the magnetization of the residual hydrodynamic modes[36]. It is found that under the influence of the imposed magnetic field, the $A_0$ value in the final steady state of the second stage is different from that in the relaxed hydrodynamic state of the first stage, indicating that the base flow is modified by the applied magnetic field[36]. Figure 4b shows the corresponding time evolution of the amplitude $A_1$ of the $m = 1$ mode. As shown by the black curve, $A_1$ in the first stage is negligibly small, as expected. Once $A_1$ is allowed to evolve in the second MHD stage, it first

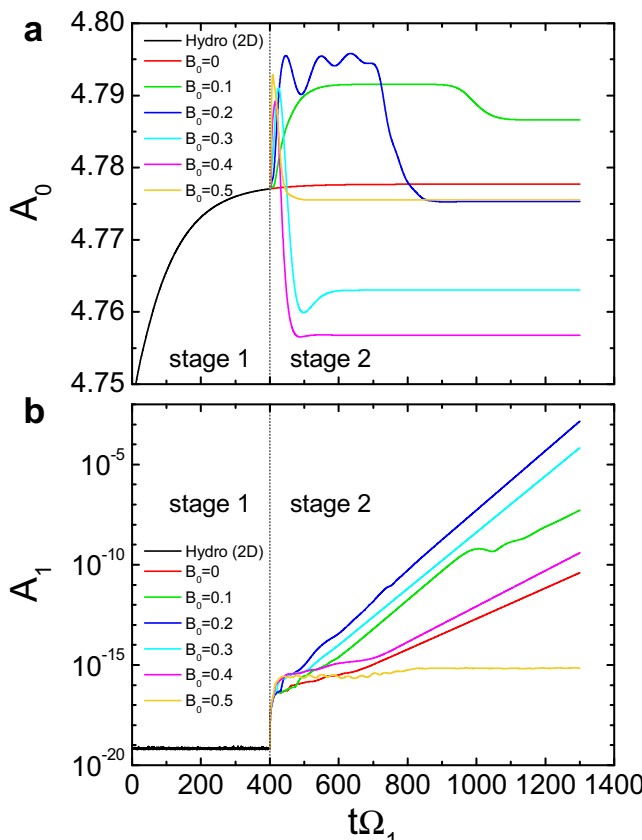

**Fig. 4 | Time evolution of 2-mode simulation.** Simulated dimensionless amplitude $A_0$ of the $m = 0$ mode (**a**) and $A_1$ of the $m = 1$ mode (**b**) in the total velocity field, as a function of the dimensionless time $t\Omega_1$ for different values of $B_0$ at fixed Rm = 4. The calculation is based on volume average over the fluid domain. Vertical dashed lines show the time boundary between the first and second stages.

rapidly increases to $\sim 3 \times 10^{-17}$ at $t\Omega_1 \simeq 410$ in all cases. Such an increase is a numerical self-adjustment to adapt to the sudden change of mode number, therefore is irrelevant to the physical flow dynamics. Similar to the amplitude of the instability identified in the experiment shown in Fig. 1d, $A_1$ grows exponentially in the final steady state of the second stage, $A_1 \sim e^{\gamma_0 t\Omega_1}$, with $\gamma_0$ being the dimensionless growth rate. The amplitude of the $m = 1$ mode in the magnetic field also has an exponential growth with a growth rate the same as that in the velocity field.

As shown by black squares in Fig. 5a, the hydrodynamic case ($B_0 = 0$) in our 2-mode simulation has a growth rate of $\gamma_0 = 0.0175 > 0$, indicating that the $m = 1$ mode is unstable to the relaxed 2D hydrodynamic state in the first stage. This is as expected because nonaxisymmetric modes are found in the relaxed hydrodynamic state of the 3D simulation. The value of $\gamma_0$ at $B_0 = 0$ thus can be taken as a benchmark, and deviations from it in other $B_0 > 0$ cases are caused by MHD effects. It is found that $\gamma_0$ from simulation is not monotonic in $B_0$, but rather is maximized at an intermediate field strength, as is characteristic of the instability observed in our experiment. For comparison, we also plot the corresponding $\gamma_0$ from experiment, which is determined by the growth of $\chi(t)$ at the beginning stage ($t \lesssim 127$ s) after the external magnetic field is imposed (see Fig. 1d). Because nonaxisymmetric hydrodynamic modes are squeezed to regions adjacent to the endcaps and thereby are absent in the bulk region of the experiment, $\gamma_0$ increases from zero with $B_0$. Except for this difference, the experiment agrees quite well with the simulation, including that $\gamma_0$ has a non-monotonic dependence on $B_0$ and becomes significantly large for $0.15 \lesssim B_0 \lesssim 0.3$. Similarly, Fig. 5b shows that at fixed $B_0 = 0.2$, $\gamma_0$ from both 2-mode simulations (black squares) and experiments (red

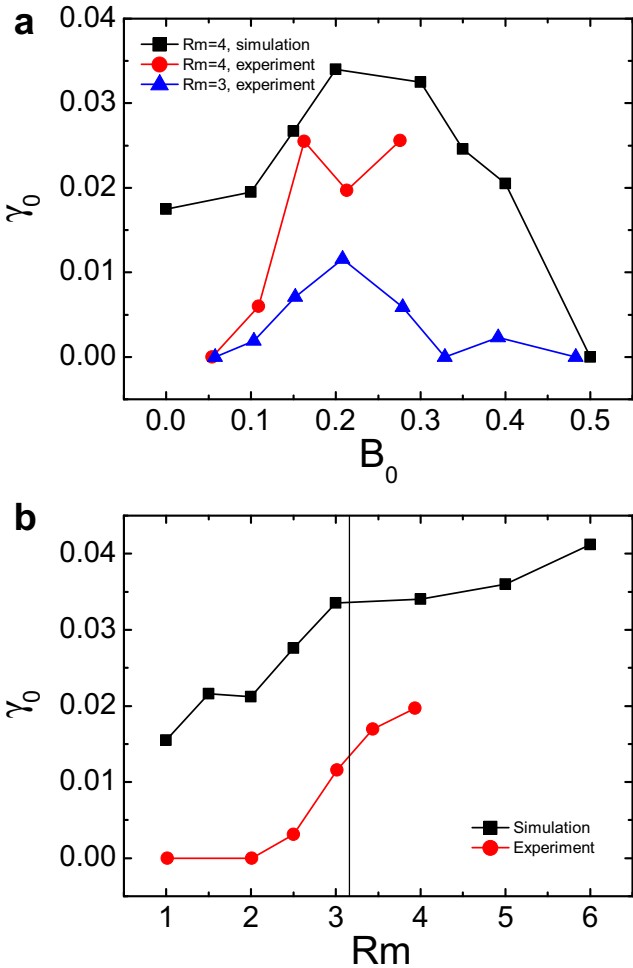

**Fig. 5 | Comparison of the $m = 1$ mode growth rate in simulation and experiment. a** Normalized growth rate $\gamma_0$ of the $m = 1$ mode from the 2-mode simulation (black squares) and the experiment (red circles and blue triangles), as a function of $B_0$ at a fixed Rm. **b** Corresponding $\gamma_0$ as a function of Rm at fixed $B_0 = 0.2$.

circles) is largely enhanced for Rm $\gtrsim 3$, consistent with the amplitude of the instability identified at saturation (see Fig. 3). Again, the positive $\gamma_0$ at Rm $\lesssim 2$ in the simulation is mainly caused by the residual hydrodynamic modes, which are not present in the experiment and thus $\gamma_0 \simeq 0$. All these findings indicate that in the presence of sufficiently large shear and a moderate axial magnetic field, the mean axisymmetric flow is driven into a new state at which nonaxisymmetric modes become linearly unstable. This mechanism exists only in bounded systems like our experiments and may not apply to a real accretion disk, where the Kepler flow is hardly affected by instabilities or turbulence in it.

Further investigations are needed to fully understand the reported instability, including its fundamental cause from first principles, saturation mechanism, transition to turbulence, and responses to different radial boundary conditions and geometric confinements. Different components of the velocity and magnetic fields inside the liquid-metal flow will be measured using Ultrasonic Doppler Velocimetry and Hall probe arrays, which help to better resolve its spatial structures and relationship to local angular momentum transport, and thus further determine its identity. It is also worth examining its possible connection to the nonaxisymmetric SMRI in a narrow-gap annular cylinder observed recently by an incompressible linear theory[47]. The possibility of nonaxisymmetric global instabilities due to unstable Alfvén continuum will also be investigated[48]. Numerical simulations

with a higher Re closer to the experimental setup should be explored, perhaps with an entropy-viscosity method in SFEMaNS[49].

## Methods

### Taylor–Couette cell

Details about the device used in this experiment have been described elsewhere[34], and here we only mention some key points. As shown in Fig. 6, the inner cylinder is composed of five Delrin rings (green) and two stainless steel caps (cyan) that compress them axially. The upper (lower) stainless steel cap has a 1 cm protruding rim on its top (bottom) side, respectively, which help to further reduce the Ekman circulation[4]. The outer cylinder is made of stainless steel. The endcaps between the two cylinders are made of 1-inch-thick silver-plated copper and split into two rings at $r_3 = 13.5$ cm. The upper and lower inner rings are bound together, while the upper and lower outer rings are bound to the outer cylinder. Driven by three independent motors, the inner cylinder, inner rings, and outer-ring-bound outer cylinder can rotate independently.

This Taylor–Couette cell has three unique features for the experiment reported here. First, the gap between the inner and outer cylinders is purposely made wide, which corresponds to a small aspect ratio Γ that helps to excite the SMRI according to theoretical predictions[28]. Second, the independently rotatable endcap rings reduce Ekman circulation that could destabilize the desired quasi-Keplerian profile[3,31]. For a conventional Taylor–Couette cell, the end-caps are either bound to the inner cylinder or to the outer cylinder, which leads to a boundary condition different from the flow profile in the bulk. As a result, the Ekman circulation is inevitably excited and highly disturbs the bulk flow[3,29], which likely overwhelms signals from the SMRI. In our cell, the inner rings rotate at an angular speed between that of the inner and outer cylinders, thereby significantly reducing the velocity discontinuity at the endcap. Consequently, the Ekman circulation in our cell is significantly reduced, allowing the flow in the bulk to approach a quasi-Keplerian profile[3,4,32,50]. Finally, the conducting copper endcaps significantly enlarge the magnetic stress within the boundary layer attached to them, which reinforces the differential rotation in the bulk flow[33–36].

### Measurement of local magnetic field

The local magnetic field is measured by high-precision Hall probes (Allegro MicroSystems, A1308 series) with an accuracy of 0.5 G. A Hall probe is a device whose output voltage is directly proportional to the magnetic field through it. As shown by arrows in Fig. 7, the Hall probes are mounted at the surface of the three Delrin rings in the middle and orientated outwards to measure the radial magnetic field $B_r$. Six Hall probes (red arrows) are placed in the upper 1/4 plane ($z = 7$ cm). Six Hall probes (blue arrows) are placed in the midplane ($z = 0$ cm) with a same azimuthal distribution. One probe (green arrow) is placed in the lower 1/4 plane ($z = -7$ cm). An Arduino-based system containing analog-to-digital converter (ADC) chips and a micro SD card is used to measure and record the voltage signals from Hall probes with a sampling rate ~ 175 Hz. The Arduino system is placed in a container along with a 9 V battery that powers it and the Hall probes. This container is bound on top of the inner cylinder and thus corotates with it.

### Characterization of nonaxisymmetric modes

We fit the time series of the radial magnetic field at different azimuths $B_r(\theta, t)$ to the azimuthal Fourier series,

$$B_r(\theta,t) = a_0(t) + \sum_{m=1}^{N} a_m(t)\cos(m\theta + \theta_m(t)), \qquad (2)$$

where $a_0(t)$, $a_m(t)$, and $\theta_m(t)$ are fitting parameters. In Eq. (2), the absolute value $|a_0(t)|$ describes the amplitude of the axisymmetric

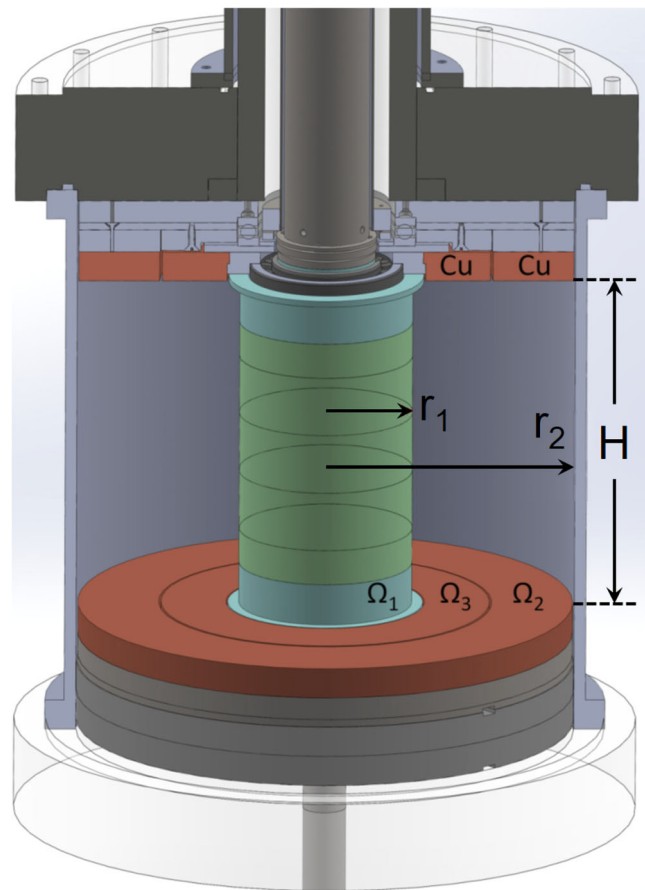

**Fig. 6 | Sketch of the Taylor–Couette cell used in the experiment.** The cell has three independently rotatable components: the inner cylinder ($\Omega_1$), outer-ring-bound outer cylinder ($\Omega_2$), and upper/lower inner rings ($\Omega_3$). This plot was created by the authors and previously published[34].

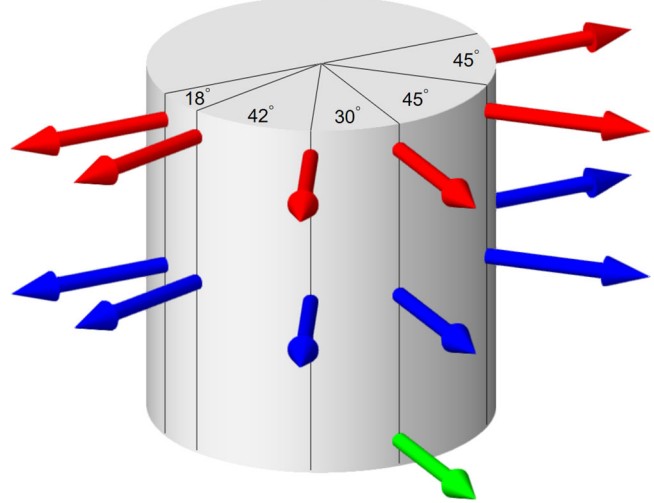

**Fig. 7 | Arrangement of Hall probes at the inner cylinder surface.** The color-coded arrows represent the position of Hall probes in the lower (green), middle (blue), and upper (red) horizontal planes. The numbers represent the azimuth difference between two adjacent probes.

mode, while $|a_m(t)|$ and $\theta_m(t)$ describe the amplitude and phase of the $m$th azimuthal mode. Since there are only six sensors in the midplane, we set $N = 2$ for data from the experiment, which leaves residuals an order of magnitude smaller than the smallest mode amplitude. For the simulations, we adopt $N = 10$, and the amplitude of the residuals relative to $B_z$ is less than $10^{-5}$. Before fitting, a bandpass filter with a pass-band $1.05f_{13} \leq f \leq 0.95f_{12}$ is applied to the experimental data in order to remove the mechanical signals (Fig. 1c). Supplementary Fig. 1 shows an example of the measured radial magnetic field variations as a function of azimuth angle $\theta$ from the experiment (Supplementary Fig. 1a) and simulation (Supplementary Fig. 1b). As shown by the solid lines, the data points are well described by Eq. (2) with $N = 2$ for experiment and $N = 10$ for simulation.

## Linear theory predictions of SMRI

The axisymmetric ($m = 0$) SMRI in our system has been extensively studied by local Wentzel–Kramers–Brillouin (WKB) analysis[27] and global linear analysis[28]. Both methods assume no change in base flow in the axial direction, therefore do not include the conducting and no-slip boundary conditions in real experiments and nonlinear simulations. The global linear analysis assumes that the base flow has a Couette profile. The base flow shear in the WKB method is the geometric mean of the Couette flow shears of the inner and outer cylinders. As shown in Fig. 8, predictions of $m = 0$ SMRI from WKB method (blue curve) and global linear analysis (green curve) are very similar, both requiring $B_z \gtrsim 5000$ G and $\Omega_1 \gtrsim 4500$ rpm (Rm $\gtrsim 9$). This is higher than the parameter space explored in current experiments and simulations, as shown by the bubbles, concentrated in the region of $B_z \lesssim 5000$ G and

$\Omega_1 \lesssim 2500$ rpm (Rm $\lesssim 5$). In particular, as shown by the magenta curve, the prediction for the $m = 1$ SMRI from global linear analysis requires even higher flow shear ($\Omega_1 \gtrsim 12,000$ rpm or equivalently Rm $\gtrsim 24$) and magnetic field strength ($B_z \gtrsim 18,000$ G).

## Characterization of the Stewartson–Shercliff layer instability

As shown in Fig. 9, the Stewartson–Shercliff layer (SSL) is a local free shear layer with $q > 2$ that originates from the junction of the inner and outer rings, where substantial shear occurs due to velocity discontinuities. The flow in SSL is Rayleigh and Kelvin–Helmholtz unstable, entailing nonaxisymmetric modes. For a fixed shear (Rm), the vertical extent of the upper and lower SSLs monotonically increases with the applied magnetic field strength, until they merge in the midplane. It has been shown that for a "split" configuration with $\Omega_3 = \Omega_1$ and insulating endcaps, the two SSLs reach the midplane once the Elsasser number

$$\Lambda = \frac{B_z^2}{\mu_0 \rho \eta (\Omega_3 - \Omega_2)} \tag{3}$$

is greater than unity, causes nonaxisymmetric fluctuations there[40,41]. Because the SSL is inductionless, Eq. (3) is valid in the small Rm limit[40]. Despite the use of conductive copper endcaps and different rotation speed ratios in our experiments, we find Eq. (3) works remarkably well for the onset of SSL instability in the midplane with Rm $\lesssim 2$ (see Fig. 3). At the same time, our simulation also shows that the two SSLs reach the midplane only for $\Lambda \gtrsim 1.6$ for all Rm values studied here, consistent with experimental results. For example, as shown in Fig. 9, at Rm = 4 the two SSLs merge in the midplane only for $B_0 \gtrsim 0.4$, which corresponds to $\Lambda \gtrsim 1.64$.

## Characterization of hydrodynamic Rayleigh instability

To excite the hydrodynamic Rayleigh instability, an angular velocity ratio $\Omega_1 : \Omega_3 : \Omega_2 = 1 : 0.507 : 0.05$ called Rayleigh unstable configuration is adopted, as Rayleigh's criteria demands $\Omega_2/\Omega_1 \leq r_1^2/r_2^2 \simeq 0.12$ for the Rayleigh instability[2]. The value of $\Omega_3$ is chosen such that the Ekman circulation is still suppressed. In the experiment, a weak $B_z$ acting as passive tracers to the hydrodynamic flow is imposed, and we use the measured $B_r(t)$ to characterize the hydrodynamic instability. Similar to the procedure discussed in the main text, a relaxed hydrodynamic flow is achieved before the imposition of $B_z$.

Supplementary Fig. 2a shows the measured normalized power spectrum $P_B(f)/B_z^2$ of $B_r(t)$ at the angular velocity ratio used in the main text. The data are obtained in the frame corotating with the inner cylinder. The vertical lines indicated the machine-induced frequency, at which the power is irrelevant to the flow dynamics. Because the imposed magnetic field is weak, MHD effects are minute and thereby the measured $B_r(t)$ represents hydrodynamic properties of the flow. As expected, the power over the whole frequency range is small, indicating that the flow is hydrodynamically stable. Supplementary Fig. 2b shows the measured power spectrum at the Rayleigh unstable

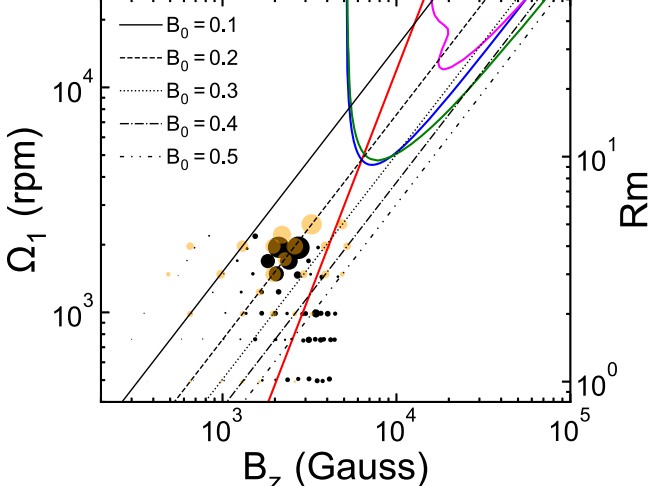

**Fig. 8 | Comparison with linear theory predictions for SMRI.** Bubble plot of the instability strength from experiments (black bubbles) and 3D simulations (orange bubbles) in the $\Omega_1$-$B_z$ plane with Rm shown on the right. The data are the same as those in Fig. 3. The blue curve represents the boundary of the $m = 0$ SMRI from the WKB analysis, which is unstable in the parameter space it encloses. The green and magenta curves represent the boundaries of the $m = 0$ and $m = 1$ SMRI from the global linear analysis, respectively. The red line represents $\Lambda = 1$ using Eq. (3).

configuration. Compared with Supplementary Fig. 2a, the power spectrum in Supplementary Fig. 2b has significant power at $f/f_1 \lesssim 0.2$. Such a low-frequency power is believed to be from the hydrodynamic Rayleigh instability, which has a frequency spectrum distinct from the instability ($0.42 \lesssim f/f_1 \lesssim 0.81$) discussed in the main text.

Using Eq. (2), mode decomposition of hydrodynamic Rayleigh instability in the azimuthal direction is also performed. Supplementary Fig. 3 shows the measured normalized mode amplitude $\langle |a_m| \rangle/B_z$ as a function of mode number $m$. Before fitting, we applied a low-pass filter with a cutoff frequency $0.2f_1$ to the measured $B_r(t)$ at different azimuths. The discretization of ADC chips gives rise to an accuracy of 0.5 G for the measured $B_r(t)$, which is about 0.1% of the imposed $B_z$ in experiments with Rayleigh unstable configuration. Segments with a variation less than 0.5 G in the filtered $B_r(t)$ are thus inaccurate and discarded. Compared with modes of instability shown in Fig. 2a, there are three main differences for the azimuthal modes of the hydrodynamic Rayleigh instability. First, the overall mode amplitudes of the hydrodynamic instability are much smaller, indicating that at least its time-varying part is quite weak. Second, unlike the instability reported in the main text that only appears for Rm $\gtrsim 3$, modes of hydrodynamic Rayleigh instability at different Rm have a similar amplitude. Finally, the dominant mode of the hydrodynamic Rayleigh instability is $m = 0$, in contrast to the instability reported in the main text that has a dominant $m = 1$ mode. This is consistent with the typical features of hydrodynamic Rayleigh instability in a Taylor−Couette flow, in which it has an evolution from axisymmetric to nonaxisymmetric[51,52]. All these differences further confirm that the instability reported in the main text is unlikely to be the hydrodynamic Rayleigh instability.

### Numerical methods and simulation setup

The numerical code used in the simulation has been described in detail elsewhere[33,35,36], and only some key points are mentioned here. As in the experiment, a cylindrical coordinate system is adopted in our three-dimensional (3D) simulation. The origin is set at the geometric center of the Taylor−Couette cell. Unit vectors in the radial, azimuthal and vertical directions are denoted by $\mathbf{e}_r$, $\mathbf{e}_\theta$, and $\mathbf{e}_z$. In the simulation, we set $r_2/r_1 = 3$, $r_3/r_1 = 2$, $H/r_1 = 4$ and the radius of the inner cylinder rim

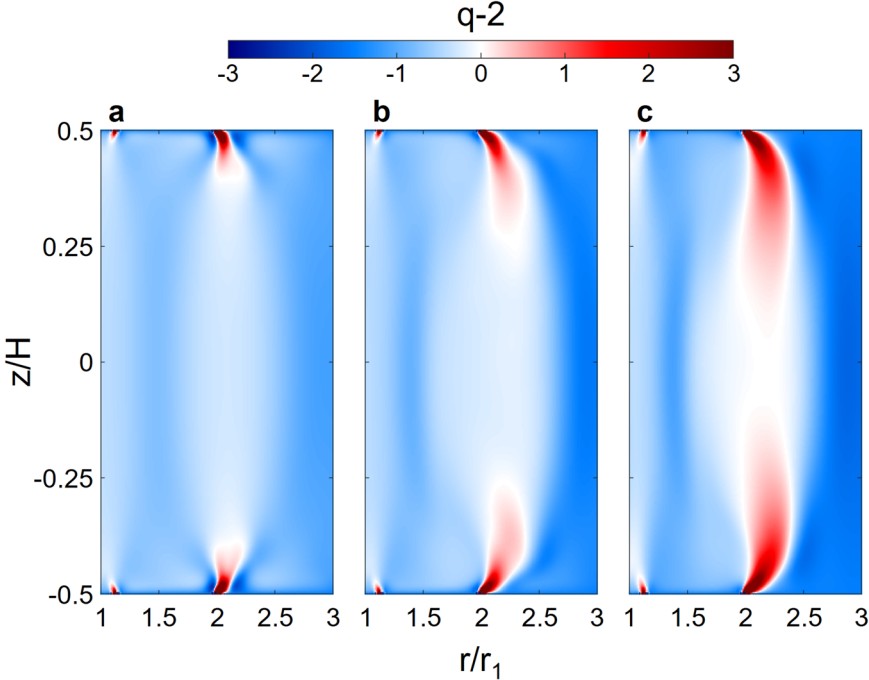

**Fig. 9 | Characterization of flow shear.** Shear profile $q - 2$ from 3D simulations at fixed Rm = 4 and different values of $B_0$ and $\Lambda$: (**a**) $B_0 = \Lambda = 0$ (hydrodynamic), (**b**) $B_0 = 0.2$ and $\Lambda = 0.41$, (**c**) $B_0 = 0.4$ and $\Lambda = 1.64$. Calculations are based on time and azimuthal averages in the hydrodynamic or MHD steady state.

$r_{rim}/r_1 = 1.15$. The length, time, velocity, magnetic field and electrical conductivity are normalized, respectively, by $r_1$, $\Omega_1^{-1}$, $\Omega_1 r_1$, $\Omega_1 r_1 \sqrt{\mu_0 \rho}$ and $\sigma_{Ga}$, where $\sigma_{Ga}$ is the electrical conductivity of galinstan. In order to mimic the experiment, the whole volume is divided into three coupled domains, including a fluid domain for galinstan, a solid domain for endcaps and a spherical vacuum domain surrounding them.

Supplementary Fig. 4a shows the mesh allocation in a quarter of the meridional plane, in which meshes with different colors belong to different domains. The fluid domain has a mesh resolution of $100 \times 200$ triangular cells. The governing dimensionless equations are the resistive MHD equation for an incompressible fluid, with

$$\frac{\partial \tilde{\mathbf{u}}}{\partial \tilde{t}} + \tilde{\mathbf{u}} \cdot \tilde{\nabla} \tilde{\mathbf{u}} = -\tilde{\nabla} \tilde{p} + \frac{1}{Re} \tilde{\nabla}^2 \tilde{\mathbf{u}} + (\tilde{\nabla} \times \tilde{\mathbf{B}}) \times \tilde{\mathbf{B}},$$

$$\frac{\partial \tilde{\mathbf{B}}}{\partial \tilde{t}} = \tilde{\nabla} \times (\tilde{\mathbf{u}} \times \tilde{\mathbf{B}}) + \frac{1}{\tilde{\sigma} Rm} \tilde{\nabla}^2 \tilde{\mathbf{B}}, \qquad (4)$$

$$\tilde{\nabla} \cdot \tilde{\mathbf{u}} = 0,$$

where $\tilde{\mathbf{u}}$, $\tilde{p}$, $\tilde{\mathbf{B}}$ and $\tilde{\sigma} = 1$ ($Rm = \Omega_1 r_1^2 \sigma_{Ga} \mu_0$) are the dimensionless velocity, pressure, magnetic field, and electrical conductivity in the fluid domain, respectively.

As shown by Supplementary Fig. 4b which is an enlarged portion of Supplementary Fig. 4a, the solid domain is further divided into two sub-domains with one for the stainless steel rim of the inner cylinder (yellow) and the other for the copper endcap (blue). In the solid domain, only the induction equation in Eq. (4) for magnetic field is evolved, where $\tilde{\sigma} = 19.4$ for copper and $\tilde{\sigma} = 0.468$ for stainless steel, respectively. The dimensionless linear velocity at the endcap boundaries is $\tilde{\mathbf{u}} = \tilde{r} \mathbf{e}_\theta$ for $1 \le \tilde{r} \le 1.15$, $\tilde{\mathbf{u}} = \Omega_3 \tilde{r}/\Omega_1 \mathbf{e}_\theta$ for $1.15 < \tilde{r} \le 2$ and $\tilde{\mathbf{u}} = \Omega_2 \tilde{r}/\Omega_1 \mathbf{e}_\theta$ for $2 < \tilde{r} \le 3$. Here $\tilde{r} \equiv r/r_1$ is the dimensionless radial position.

The vacuum domain has a radius of $20 r_1$. By introducing a scalar potential for the magnetic field with $\tilde{\mathbf{B}} \equiv \tilde{\nabla} \phi$, the governing equation in the vacuum domain is

$$\tilde{\nabla}^2 \phi = 0. \qquad (5)$$

Equation (5) is a Laplace equation so its solution is uniquely determined by $\phi$ at the boundary of the domain. The boundary conditions for Eqs. (4)–(5) are the following: No-slip boundary conditions are applied at the fluid-solid interface; $\tilde{\mathbf{u}} = 1\mathbf{e}_\theta$ at $\tilde{r} = 1$ and $\tilde{\mathbf{u}} = \Omega_2/\Omega_1 \mathbf{e}_\theta$ at $\tilde{r} = 3$ are adopted in the fluid domain with insulating boundary conditions; The external magnetic field is introduced by setting the scalar potential $\phi = B_0 \tilde{z}$ at the outer boundary of the vacuum domain. In simulation, the hydrodynamic Reynolds number is fixed at $Re \equiv \Omega_1 r_1^2/\nu = 1000$, while the $Rm$ and $B_0$ are varied over their experimentally accessible ranges. The angular speed ratio in simulation is fixed at $\Omega_1 : \Omega_3 : \Omega_2 = 1 : 0.58 : 0.19$, the same as the experiment.

The 3D simulations presented here are more faithful to the current experiment than our past numerical efforts. Some of the latter were 2D and axisymmetric[33,36], and therefore could not exhibit the experimentally observed nonaxisymmetric modes. In one of our previous 3D simulations[42], pseudovacuum boundary conditions were adopted for the vertical boundaries, obviating the strong coupling between the liquid-metal flow and conducting endcaps in the experiment (at that time, the experiment had insulating endcaps). In the 3D simulations of ref. 35, the external magnetic field was imposed at the beginning of the first hydrodynamic stage, and the simulation only lasted for a short period of time: long enough for the axisymmetric mode to saturate but not for nonaxisymmetric modes to grow. We have continued to run that simulation for a longer period of time and found that eventually nonaxisymmetric modes became significant. Compared with all our previous simulation, the simulation provided in

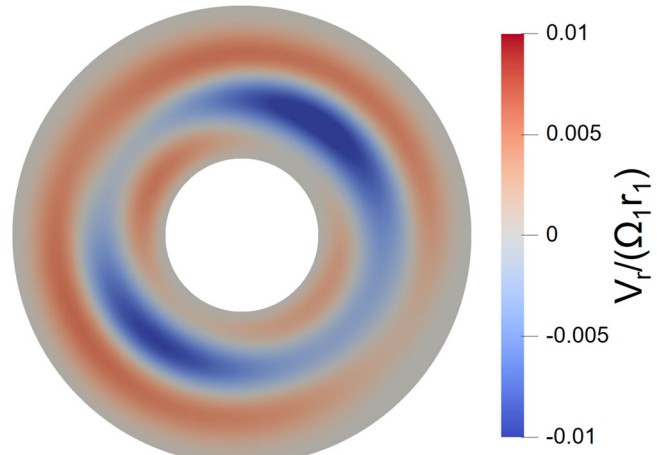

**Fig. 10 | Residual hydrodynamic modes in simulation.** Simulated normalized radial velocity profile $v_r/(\Omega_1 r_1)$ in the midplane. The red and blue regions represent outward and inward flows, respectively.

this work has the best consistency with our experiment. This includes the 3D domains, the conducting boundary conditions and the way to introduce the external magnetic field.

### Flow characterization in 3D simulation

Figure 10 shows a representative radial velocity profile $v_r/(\Omega_1 r_1)$ in the relaxed hydrodynamic state of 3D simulation. It is found that there are nonaxisymmetric modes dominated by $m = 2$ in $v_r/(\Omega_1 r_1)$, which we believe are the main cause of the positive growth rate of the $m = 1$ mode at $B_0$ or low $Rm$ in our 2-mode simulation (see Fig. 5). Similar nonaxisymmetric structures are also found in the corresponding vertical velocity profile. Such structures only exist in simulation with low $Re$ ($Re = 1000$), but are completely suppressed in our high $Re$ ($Re \sim 10^6$) experiments[39]. In addition, the frequencies of the nonaxisymmetric hydrodynamic modes are also different from those of the instability reported in the main text, indicating that the corresponding mechanisms are different.

Supplementary Fig. 5a shows the azimuthally and vertically averaged angular speed profile $\Omega(r)$ in the bulk region at $Rm = 6$ and $B_0 = 0.2$. The origin of time ($t = 0$) in these plots coincides with the moment when the magnetic field is imposed in the second MHD stage. Three representative epochs are examined: $t\Omega_1 = 0$ corresponds to the relaxed hydrodynamic state, $t\Omega_1 = 30$ corresponds to the moment with growing nonaxisymmetric MHD modes, and $t\Omega_1 = 100$ corresponds to the saturated MHD state. Although the difference is small, the angular speed profile is indeed modified by the imposed magnetic field. Supplementary Fig. 5b shows the corresponding time evolution of $q$ profile, which also changes after applying the magnetic field, thus confirming the modification of the base flow.

### Azimuthal magnetic field

The rotating conductive endcaps drag the imposed magnetic field lines that are static in the lab frame, inducing an azimuthal magnetic field $B_\phi$ in the liquid-metal flow. As shown in Supplementary Fig. 6, the induced normalized azimuthal magnetic field $B_\phi/B_z$ in our 3D simulation is mainly concentrated in the region close to the endcaps. This is as expected since the endcaps have the most influence on the flow in contact with them. Due to the system's reflection symmetry about the midplane, $B_\phi$ in the upper and lower halfplanes changes sign. Overall, $B_\phi/B_z$ is small for all $Rm$ and $B_0$ studied here, less than 0.15 in the region close to the endcap and less than 0.05 in the bulk region. Similar results were obtained for all $Rm$ values studied here.

## Data availability

Source data of plots in the main text are deposited in the DataSpace at Princeton University that is available to the public via https://dataspace.princeton.edu/handle/88435/dsp01x920g025r. All other data that support the plots within this paper and other findings of this study are available from the author upon reasonable request.

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

## Acknowledgements

This research was supported by U.S. DoE (Contract No. DE-AC02-09CH11466), NASA (Grant No. NNH15AB25I), NSF (Grant No. AST-2108871), and the Max-Planck-Princeton Center for Plasma Physics (MPPC). This publication was supported by the Princeton University Library Open Access Fund. E.G. and H.J. acknowledge support by S. Prager and Princeton University. H.J. acknowledges the contribution by E. Schartman of Nova Photonics, Inc. funded by NSF.

## Author contributions

H.J. and J.G. initiated the research. Y.W. and E.G. performed the experiments. K.C. participated in the early experiments. F.E. updated the simulation setup with the latest version of the SFEMaNS code. Y.W. performed the simulations with the help of F.E. and H.W. Y.W. analyzed the data and prepared the figures with the help of E.G., F.E., J.G., and H.J. Y.W. drafted the manuscript. Y.W., E.G., F.E., J.G., and H.J. discussed the results and revised the manuscript together.

## Competing interests

The authors declare no competing interests.
