## [Peer Review File · Nature Communications]

Identification of a non-axisymmetric mode in laboratory experiments searching for standard magnetorotational instabilityEditorial Note: This manuscript has been previously reviewed at another journal that is not operating a transparent peer review scheme. This document only contains reviewer comments and rebuttal letters for versions considered at *Nature Communications*.

REVIEWER COMMENTS

Reviewer #1 (Remarks to the Author):

I applaud the authors for having brought the third iteration of this paper into a form which is nearly publishable. They present now an interesting MHD effect - occurring at significant magnetic Reynolds and Lundquist numbers - without prematurely ascribing it to the SMRI whose critical parameters are still far away from the experimentally achieved ones, as now clearly shown in Fig. 9, both for $m=0$ and, even more strikingly, for $m=1$. That said, I have the impression that at quite a number of occasions the former "bias" in favour of SMRI, and against any competitive, and more obvious, explanation is still "shining through". This applies, in particular, to the counterarguments against a Rayleigh-unstable flow on page 10 (lines 563-579). This argumentation is, for my taste, quite ill-founded, given the clear existence of $q>2$ areas with significant size shown in Fig. 10. Since those areas are quite thin, the arguments of page 10, which apply to the Rayleigh-unstable setting of the WHOLE flow, are not really conclusive here, in particular regarding the distinction between $m=0$ and $m=1$. In this respect, I'm also a bit irritated by the strong wording in the abstract ("Further analysis also EXCLUDES the possibility..."). I would have much preferred a more cautious formulation here such as "Further

analysis suggests that it is not the...", say...

There are also some remnants of former claims for SMRI, e.g. on lines 68-71, lines 276-277, lines 435-426, lines 646-648, lines 674-676, which authors should work over in the next revision.

Apart from that, I have only a few minor comments:

- line 54: viscous -- viscously ??

- line 181: amplitude -- amplitudes

- line 265: "Even conductive" - "Even as conductive"

- lines 370/371: component - components

- line 443: "an sampling" -> "a sampling"

- lines 612/613: I guess the two values for $\tilde{\sigma}$ for copper and stainless steel must be interchanged

Reviewer #4 (Remarks to the Author):

See attached report

**Report on —
'Identification of a novel non-axisymmetric mode in laboratory experiments searching for standard magnetorotational instability'**

This work describes experimental results obtained from a magnetised Couette flow between two rotating flows with separately rotating end caps. The object was to validate instabilities believed to occur in astrophysical discs and enable angular momentum transport and accretion to occur. However, responding to referee's reports the authors accept that this has not been achieved with adequate clarity and restrict their claim to that of the discovery of a new MHD instability. The authors have worked hard to show that this is indeed present and in this context the work is publishable (but connection to astrophysics remains questionable see below)

Interpretation of the results obtained is notoriously difficult on account of the interference of boundaries on the small scale experiments. These produce other kinds of instabilities and can contaminate any normal modes discovered. The referees have correctly highlighted these issues.

For astrophysical application modes that do not depend on boundaries are the focus of attention (as noted by referee 2). For a smooth self-similar disc non-axisymmetric modes are expected to exhibit transient growth only. A genuine growing normal mode that does not depend on the influence of boundaries requires internal structural features.

This is probably why the authors considered the possibility of axisymmetric modes producing such features that could induce a non axisymmetric instability of the type they see. However, they seem later to have abandoned this approach and only propose a new nonaxisymmetric MHD instability with unclear connection to astrophysical phenomena.

Thus, as the title of the manuscript indicates, what the authors have done is to provide a step in the direction of finding confirmation of the standard MRI. In this context the work is publishable but it should be noted that this is not in a position to be carried over to astrophysics at present.

Response to Referee 1:

The authors wish to thank the referee for his/her positive recommendation and constructive suggestions on this work. The following are the changes made in response to each of the referee's comments (marked in red).

COMMENTS FOR THE AUTHOR:

Reviewer #1:

I applaud the authors for having brought the third iteration of this paper into a form which is nearly publishable. They present now an interesting MHD effect - occurring at significant magnetic Reynolds and Lundquist numbers - without prematurely ascribing it to the SMRI whose critical parameters are still far away from the experimentally achieved ones, as now clearly shown in Fig. 9, both for $m=0$ and, even more strikingly, for $m=1$. That said, I have the impression that at quite a number of occasions the former "bias" in favour of SMRI, and against any competitive, and more obvious, explanation is still "shining through". This applies, in particular, to the counterarguments against a Rayleigh-unstable flow on page 10 (lines 563-579). This argumentation is, for my taste, quite ill-founded, given the clear existence of $q>2$ areas with significant size shown in Fig. 10. Since those areas are quite thin, the arguments of page 10, which apply to the Rayleigh-unstable setting of the WHOLE flow, are not really conclusive here, in particular regarding the distinction between $m=0$ and $m=1$.

In the subsection "Characterization of hydrodynamic Rayleigh stability" in Methods, we focused on the *hydrodynamic* Rayleigh instability, in which only a very weak magnetic field acting as passive tracers to the hydrodynamic flow is applied. The hydrodynamic Rayleigh instability is axisymmetric at its onset. On the other hand, the enhanced $q>2$ regions [Stewartson-Shercliff layer (SSL) instability] shown in Fig. 10 are caused by a strong magnetic field and is therefore more of an MHD effect. The SSL instability is nonaxisymmetric and the $m=0$ mode could be stabilized by the excessive magnetic field. So in principle these are two different types of instabilities (hydrodynamic Rayleigh instability and SSL instability), which should be treated separately as we have done in the manuscript.

In this respect, I'm also a bit irritated by the strong wording in the abstract ("Further analysis also EXCLUDES the possibility..."). I would have much preferred a more cautious formulation here such as "Further analysis suggests that it is not the...", say...

Following the referee's suggestion and in conjunction with the discussion above, we have revised this sentence in the abstract to read "Further analysis also suggests that it is unlikely to be the hydrodynamic Rayleigh instability or the Stewartson-Shercliff layer instability". We have also revised the conclusive sentence in the Methods accordingly (lines 568-570).

There are also some remnants of former claims for SMRI, e.g. on lines 68-71, lines 276-277, lines 435-426, lines 646-648, lines 674-676, which authors should work over in the next revision.

In the revised manuscript, we have removed the claim of SMRI in the text as pointed by the referee (lines 68-70, 274-279, 415-418, 637, 661-665).

Apart from that, I have only a few minor comments:

line 54: viscous -- viscously ??

Correction has been made as suggested.

line 181: amplitude -- amplitudes

The subject of the sentence here is "the m -th azimuthal Fourier mode", which is singular, so

“amplitude” is correct.

line 265: "Even conductive" - "Even as conductive"

We thank the referee for pointing out the grammatical error here. To avoid ambiguity, we changed the beginning of the sentence to “Even with conductive endcaps, ...” (lines 264).

lines 370/371: component - components

Correction has been made as suggested.

line 443: "an sampling" -> "a sampling"

Correction has been made as suggested.

lines 612/613: I guess the two values for $\tilde{\sigma}$ for copper and stainless steel must be interchanged.

Correction has been made as suggested.

Response to Referee 2:

The authors wish to thank the referee for his/her positive recommendation and constructive suggestions on this work. The following are the changes made in response to referee's comments (marked in red).

Referee: 2

Comments to the Author

Report on "Identification of a novel non-axisymmetric mode in laboratory experiments searching for standard magnetorotational instability"

This work describes experimental results obtained from a magnetised Couette flow between two rotating flows with separately rotating end caps. The object was to validate instabilities believed to occur in astrophysical discs and enable angular momentum transport and accretion to occur. However, responding to referee's reports the authors accept that this has not been achieved with adequate clarity and restrict their claim to that of the discovery of a new MHD instability. The authors have worked hard to show that this is indeed present and in this context the work is publishable (but connection to astrophysics remains questionable see below)

Interpretation of the results obtained is notoriously difficult on account of the interference of boundaries on the small scale experiments. These produce other kinds of instabilities and can contaminate any normal modes discovered. The referees have correctly highlighted these issues.

For astrophysical application modes that do not depend on boundaries are the focus of attention (as noted by referee 2). For a smooth self-similar disc, non-axisymmetric modes are expected to exhibit transient growth only. A genuine growing normal mode that does not depend on the influence of boundaries requires internal structural features.

This is probably why the authors considered the possibility of axisymmetric modes producing such features that could induce a nonaxisymmetric instability of the type they see. However, they seem later to have abandoned this approach and only propose a new nonaxisymmetric MHD instability with unclear connection to astrophysical phenomena.

Thus, as the title of the manuscript indicates, what the authors have done is to provide a step in the direction of finding confirmation of the standard MRI. In this context the work is publishable, but it should be noted that this is not in a position to be carried over to astrophysics at present.

We agree with the referee that our current understanding of the nonaxisymmetric instability observed in our system cannot be applied to astrophysical accretion disks, which was already mentioned at the end of the penultimate paragraph in the previous version of our manuscript (lines 356-359 in the latest version). The text (lines 249-250, 276-277 in the previous version) about the connection between our findings and astrophysical disks has also been removed.

REVIEWERS' COMMENTS

Reviewer #1 (Remarks to the Author):

The authors have complied with all my suggestions. I recommend publication of the paper in the present form.